# Crystallographic and Theoretical Study of Osme Bonds in Nitrido-Osmium(VI) Complexes

Rosa M. Gomila 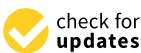 and Antonio Frontera *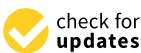

Department of Chemistry, Universitat de les Illes Balears, Crta de Valldemossa km 7.5, 07122 Palma de Mallorca, Spain
* Correspondence: toni.frontera@uib.es

**Abstract:** Osme bonds have been recently defined as the attractive interaction between an element of group 8 acting as an electrophile and any atom or group of atoms acting as a nucleophile. To date, the known examples of osme bonds in X-ray structures involve mostly the highly reactive $OsO_4$ and amines and amine oxides. In this work, evidence supporting the existence of osme bonds in osmium(VI) derivatives is reported. In particular, nitrido-osmium(VI) complexes that present square-pyramidal geometries are well disposed to participate in osme bonds opposite to the Os≡N bond. By using a combination of experimental and theoretical results, the existence and importance of this new class of σ-hole interactions is demonstrated in the solid state of several nitrido-osmium(VI) derivatives.

**Keywords:** σ-hole interactions; crystal engineering; DFT calculations; osmium

## 1. Introduction

Noncovalent interactions involving p-block elements from groups 18 to 13 as electron acceptors, namely aerogen, halogen, chalcogen, pnictogen, tetrel and triel bonds, respectively, are, nowadays, commonly used by synthetic chemists (for catalytic purposes), supramolecular chemists and crystal engineers [1–21]. These interactions have been rationalized using the σ-hole and π-hole concepts proposed by Politzer and coworkers [22].

More recently, theoretical and crystallographic scientists are expanding the σ-hole concept to the d-block of elements (groups 3–10) and post-transition metals (groups 11 and 12). In fact, several noncovalent interactions were defined by combining theoretical and experimental evidence, which are regium (or coinage) [23–25] and spodium bonding [26] interactions (SpB) for post-transition elements of groups 11 and 12, respectively. In these interactions, the metal centers act as Lewis acids, and these names are used to differentiate noncovalent interactions from coordination bonds, which have a more covalent character. Similarly, the terms "wolfium bond" [27] and "matere bond" [28] were used to define the net attractive interaction between elements from groups 6 and 7, respectively, acting as Lewis acids and any electron donor.

Experimental and theoretical evidence of σ-hole interactions in adducts between electron-rich atoms (pyridines and pyridine N-oxide derivatives) and osmium tetraoxide has been recently reported [29]. The term "osme bond" (OmB) has been proposed and used [30] for defining the noncovalent interaction wherein any element belonging to group 8 has the role of the electrophile. To date, the experimental evidence of osme bonds is limited to osmium(VIII) derivatives. Herein, we report experimental and theoretical evidence of the formation of osme bonds in nitrido-osmium(VI) complexes (see Scheme 1), which is unprecedented in the literature. These high-valence nitrido-osmium(VI) complexes have been known for a long time [31], and they exhibit interesting photochemical [32–34] and electrochemical [35,36] properties. However, they have not been used to date to analyze their ability to establish osme bonds opposite to the Os≡N triple bond, as far as

our knowledge extends. A search in the Cambridge Structural Database (CSD) [37] has been performed, showing a few X-ray structures exhibiting OmBs that are important for determining their crystal packing. Moreover, theoretical density functional theory (DFT) calculations further support the formation of OmB in nitrido-osmium(VI) derivatives and the noncovalent nature of the interaction, which was analyzed using several computational tools, as explained in the following sections.

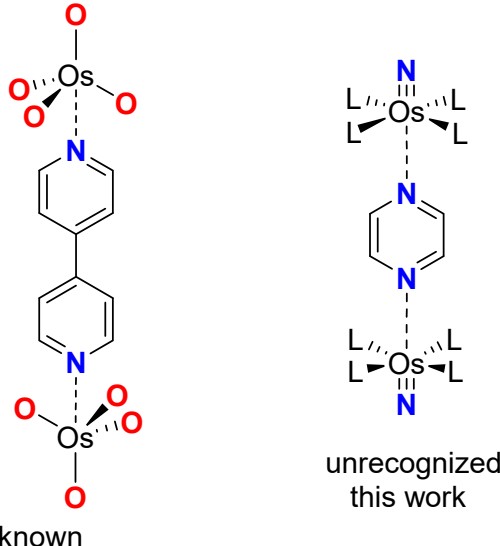

**Scheme 1.** Osme bonds previously described in the literature (**left**) [29] and the new ones proposed in this work (**right**) using nitro-osmium(VI) derivatives.

## 2. Results and Discussion

### 2.1. CSD Search

We have explored the entire CSD and found 25 X-ray structures containing the nitrido-Os(VI) fragment, which are listed in Table 1. Only four structures form directional Os···X osme bonds, as represented in Figure 1 and indicated in italics in Table 1. This is likely due to the fact that, in many structures, the [OsNL$_4$] moiety is negatively charged and, consequently, the interaction of this unit with electron-rich species is not favored.

**Table 1.** CSD reference codes of the nitrido-osmium(VI) derivatives present in the database. Those exhibiting osme bonds are shown in italics.

| Reference Codes | | |
|---|---|---|
| BAGRIK | LIFJUF | SEMHEA |
| *FEJZUP* | MIZWAT | SEMHIE |
| FEKBEC | MOQQUE | *VECMUL* |
| FOCWOK | NOGZEO | WAKGUK |
| FUTNAK | NOGZIS | WAKHAR |
| HUZYUW | *NOGZOY* | YACDIP |
| HUZZAD | QADJOV | YUNYUY |
| HUZZEH | QADJUB | |
| *JOMKAX* | QUFYAU | |

The first selected example (refcode FEJZUP) corresponds to a neutral complex where the organic ligand is trianionic, 1-(4-tert-Butylpyridine-2-carboxamido)-4,5-dichloro-2-(2-hydroxybenzamido)benzene (H$_3$L), thus, forming a complex with the formula OsNL. In solid state, it forms self-assembled dimers, where the phenoxi O atom of one monomer is located opposite to the N≡Os bond of the other monomer, and vice versa, thus, forming two symmetrically equivalent OmBs (Os···O distance: 2.878 Å). This distance is longer than

the sum of the covalent radii $\Sigma R_{cov}$(Os+O) = 2.00 Å, thus, suggesting a strong noncovalent character. A remarkable example is shown in Figure 2b, where a pyrazine molecule cocrystallizes with two molecules of tetrachloro-nitrido-osmium(VI) forming 1:2 adducts. The Os···N interactions involving the pyrazine nitrogen atoms are longer (2.518 Å) than the $\Sigma R_{cov}$(Os+N) = 2.15 Å but remarkably shorter than the sum of the van der Waals radii of the involved atoms $\Sigma R_{vdw}$(Os+N) = 3.55 Å, thus, evidencing that the self-assembly of the cocrystals is largely driven by both OmBs. The interactions are quite directional (N≡Os···N angles = 177°). It is also worth emphasizing that the [OsNCl$_4$]$^-$ units are negatively charged. For clarity, the counterions (tetrabutylammonium) are not shown. The bond distances of these OmBs are slightly longer than those previously reported in similar adducts of OsO$_4$ and bipyridine [29,30]. An intramolecular version of the OmB is observed in the NOGZOY structure, where the Os···N distance opposite to the Os≡N bond is longer than the other two Os–N distances in the equatorial plane (see Figure 1c) but much shorter than the $\Sigma R_{vdw}$(Os+N) = 3.55 Å. Therefore, in this particular case, a partial covalent character of the OmB is envisaged. A similar situation is observed for pentacyano-nitrido-osmium(VI), refcode VECMUL, where the apical cyano groups exhibit a longer Os···C distance (2.364 Å) than the other four Os···C equatorial distances that range from 2.066 Å to 2.093 Å, almost identical to the $\Sigma R_{cov}$(Os+C) = 2.10 Å. For the last two examples (NOGZOY and VECNUL), the OmB distances are similar to those previously reported for OsO$_4$ [29].

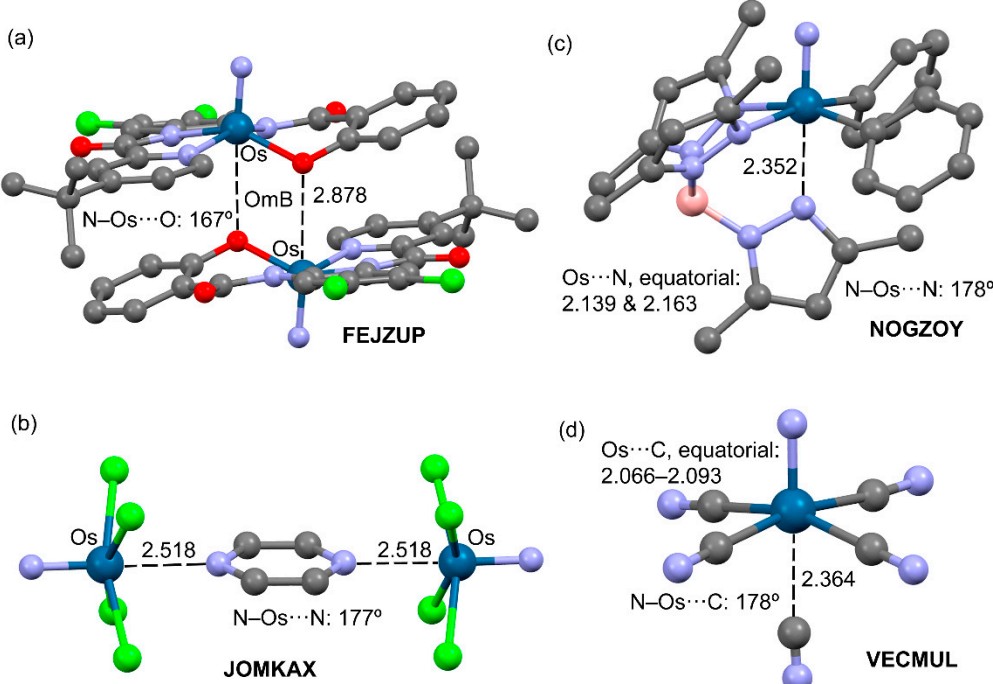

**Figure 1.** (**a**) Partial view of the X-ray structure with refcode FEJZUP. Distances are in Å. H atoms are omitted for clarity; (**b**) partial view of the X-ray structure with refcode JOMKAX. Distances are in Å. H atoms and counter-ions are omitted for clarity; (**c**) partial view of the X-ray structure with refcode NOGZOY. Distances are in Å. H atoms are omitted for clarity; (**d**) partial view of the X-ray structure with refcode VECMUL. Distances are in Å. H atoms and counter-ions are omitted for clarity.

## 2.2. MEP Surface Anaysis

Inspired by the JOMKAX structure, the tetramethyl ammonium salt of tetrachloro-nitrido-osmium(VI) (**1**) has been used as a theoretical model to investigate the existence of a σ-hole on the extension of the N≡Os bond. Figure 2 shows the MEP surfaces of compound **1** using two different orientations and MEP scales. The MEP maximum is, as expected, located at the cationic part of the salt (+79.1 kcal/mol), and the minimum MEP is located

along the bisector of the Cl–Os–Cl angle (−56.5 kcal/mol). Remarkably, the MEP surface reveals the presence of a σ-hole opposite to the N≡Os bond (+11.0 kcal/mol), which is adequate for interacting with electron-rich atoms. It is interesting to highlight the positive MEP value, taking into consideration the global negative charge of the $[OsNCl_4]^-$ unit. This analysis is useful to understand the formation of the JOMKAX cocrystal and the strong directionality of the interaction.

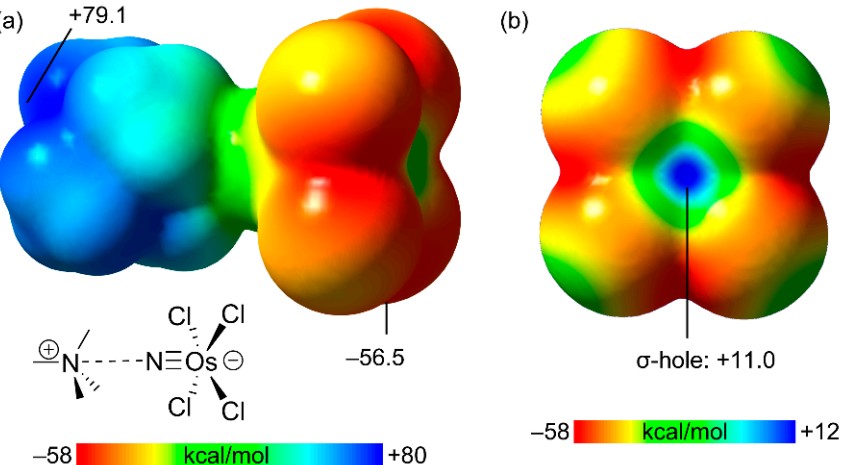

**Figure 2.** MEP surface of compound **1** using perspective (**a**) and "on-top" (**b**) representations. The MEP values at some points on the surfaces are given in kcal/mol. Level of theory: PBE0-D3/def2-TZVP. Isosurface: 0.002 a.u.

### 2.3. Energetic and Geometric Analyses

The energetic and geometric features of some model osme-bonded complexes were analyzed in this section. Three neutral Lewis bases and two anionic electron donors (see Scheme 2) were selected to investigate the nature and strength of the OmBs.

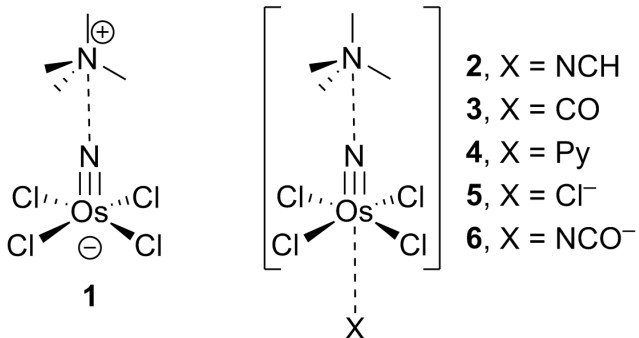

**2**, X = NCH
**3**, X = CO
**4**, X = Py
**5**, X = Cl⁻
**6**, X = NCO⁻

**Scheme 2.** Compound **1** and complexes **2**–**6** used in this work.

Table 2 gathers the interaction energies and equilibrium distances for complexes **2**–**6**. Moreover, the N≡Os–Cl angle values are also summarized in Table 2 as a measure of the deformation of the tetrachloro-nitrido-osmium(VI) unit upon complexation. It can be observed that, in all cases, the interaction energies are moderately strong, varying from −3.33 to −13.82 kcal/mol. The equilibrium distances of the complexes with neutral Lewis bases are clearly longer than the $\Sigma R_{cov}$ (values in parenthesis in Table 2), thus, disclosing a noncovalent character. This is confirmed by the low deformation of the square-pyramidal geometry of the Os atom since the N≡Os–Cl angles in complexes **2**–**4** are similar to the angle observed in **1** (104.4°). For the anionic donors, the equilibrium distances are almost identical to the $\Sigma R_{cov}$ values, thus, suggesting the formation of covalent bonds. Moreover,

the geometry of the Os atom changes to an almost perfect square-bipyramid geometry with N≡Os–Cl angles close to 90°.

**Table 2.** Interaction energies (ΔE, kcal/mol), equilibrium distances (d, Å), N≡Os–Cl angle (α, °) and electron density at the bond critical points (ρ, a.u.) of complexes **2–6**.

| Complex | ΔE | d [1] | α [2] | ρ |
|:---:|:---:|:---:|:---:|:---:|
| **2** | −3.10 | 2.523 (2.15) | 100.488 | 0.038 |
| **3** | −5.04 | 2.604 (2.21) | 101.565 | 0.037 |
| **4** | −13.82 | 2.555 (2.15) | 100.503 | 0.042 |
| **5** | −9.11 | 2.452 (1.46) | 92.227 | 0.067 |
| **6** | −3.33 | 2.095 (2.00) | 94.290 | 0.082 |

[1] Measure from the osmium to the electron-rich atom. [2] The $\Sigma R_{cov}$ is indicated in parenthesis. The N≡Os–Cl angle in **1** is 104.4°.

### 2.4. Combined QTAIM/NCIplot Analysis

In order to characterize the osme bonds in complexes **2–6**, the quantum theory of atoms in molecules (QTAIM) and the noncovalent interaction plot (NCIplot) analyses were combined in the same representation since they are useful to reveal noncovalent interactions in real space. In these representations, blue and green colors are used for strong and weak interactions, respectively, and red and yellow colors are used for strong and weak repulsive interactions, respectively. Figure 3 shows the QTAIM analyses for all compounds, evidencing that, in all cases, the OmB is characterized by a bond critical point (CP, represented as a small red sphere) and a bond path (represented by an orange line) that connects the Os atom to the electron-rich atom of the Lewis base/anion. The density values at the bond CPs are summarized in Table 2, showing larger values for the anionic complexes in line with the shorter distances. In these representations, only the bond CPs and the bond paths connecting the [OsNCl$_4$]$^-$ anionic part of the salt and the electron donors are shown.

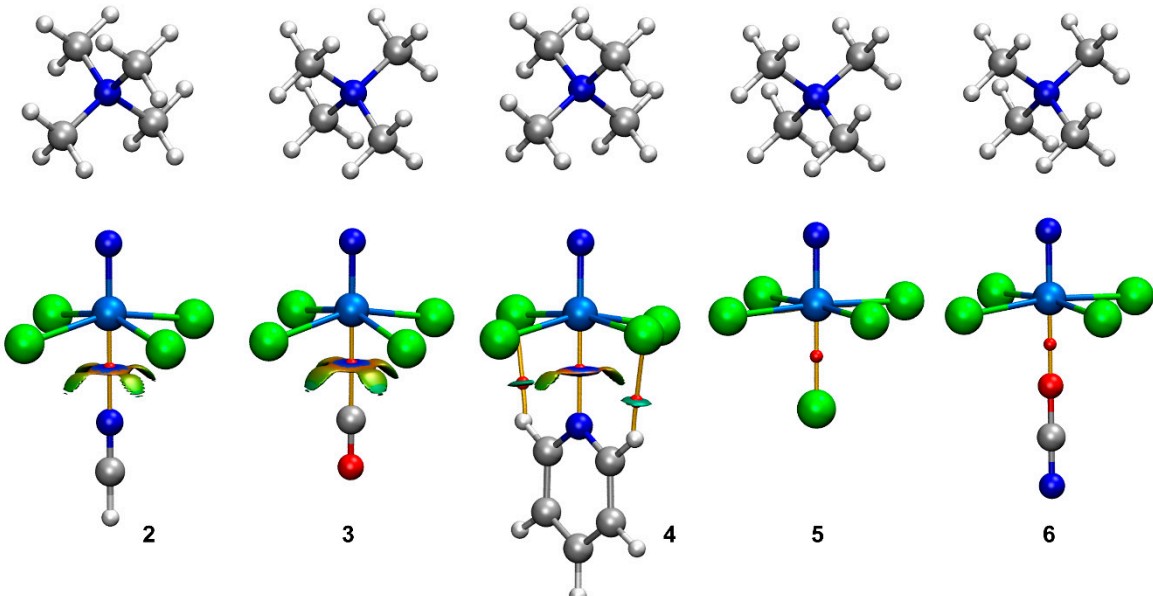

**Figure 3.** QTAIM/NCIPlot analyses of the OmB complexes **2–6** at the PBE0-D3/def2-TZVP. Only intermolecular contacts between the anionic unit and the electron donors are represented by bond CPs and RDG isosurfaces. See computational methods for the NCIplot settings.

The NCIplot analysis shows the presence of reduced density gradient (RDG) isosurfaces for the complexes with the neutral electron donors (**2–4**). For the anionic complexes, no RDG isosurfaces were obtained below the cut-off of 0.05 a.u., used for the electron

density in these plots, thus, further confirming the covalent nature of the interaction for the anionic donors. For complexes **2**–**4**, a blue RDG isosurface (meaning attractive interaction) is found coincident with the location of the bond CPs. Additional yellowish isosurfaces are located between the Cl atoms and the lone pair donor atom, thus, suggesting some repulsion with the negative belts of the chlorido ligands. In the case of pyridine, the QTAIM/NCIplot analyses reveal the existence of two ancillary C–H⋯Cl H-bonding interactions that are characterized by the corresponding bond CPs, bond paths and bluish RDG surfaces interconnecting the H and Cl atoms. The existence of the C–H⋯Cl interactions explains the larger binding energy obtained for this complex (see Table 2).

### 2.5. NBO Analysis

A differentiating feature of the σ-hole bonding with respect to coordination bonds is the existence of the typical LP→σ* donor–acceptor interaction. A convenient strategy to analyze such an orbital effect is the natural bond orbital (NBO) method and, in particular, the second-order perturbation analysis (see theoretical methods below). The analysis was been performed on the neutral donors, since they establish non-covalent OmBs. Figure 4 represents the donor and acceptor orbitals and the stabilization energies due to the LP→σ*(Os–N) orbital interaction ($E^{(2)}$ values). In all cases, the LP at the electron-rich atom points to the antibonding σ*(Os–N) orbital that is composed of the $dz^2$ atomic orbital of Os and the sp orbital of N. The $E^{(2)}$ values are large in all cases due to a significant overlap of the donor–acceptor orbitals, in agreement with the short distances. Such large orbital donor–acceptor contributions likely compensate for the electrostatic repulsion between the anionic tetrachloro-nitrido-osmium(VI) moiety and the lone pair. This NBO analysis strongly supports the σ-hole nature of the OmB contacts in complexes **2**–**4**.

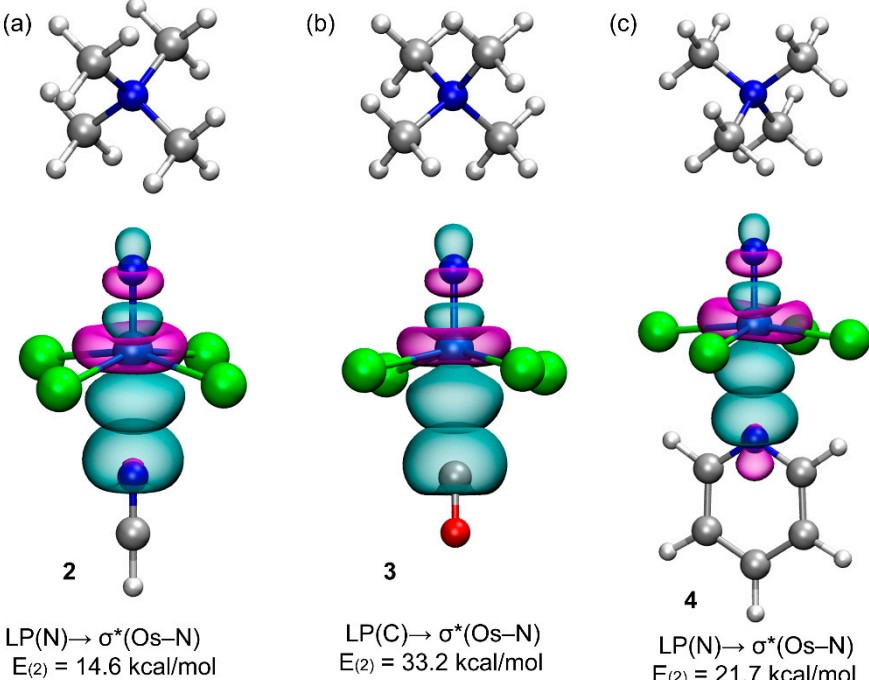

(a)

(b)

(c)

**2**

LP(N)→ σ*(Os–N)
$E_{(2)}$ = 14.6 kcal/mol

**3**

LP(C)→ σ*(Os–N)
$E_{(2)}$ = 33.2 kcal/mol

**4**

LP(N)→ σ*(Os–N)
$E_{(2)}$ = 21.7 kcal/mol

**Figure 4.** Representation of the NBOs corresponding to the LP→σ* donor–acceptor interactions in the OmB complexes **2** (**a**), **3** (**b**) and **4** (**c**). The isosurface used for the MOs is 0.008 a.u.

### 2.6. Electron Density (ED) vs. Electrostatic Potential (ESP) Analysis

In order to further analyze the nature of the Os⋯N contact in compound **4** (also as a model of the X-ray structure JOMKAX), the order of electron density (ED; $\rho(r)_{min}$) and the electrostatic potential (ESP; $\varphi(r)_{min}$) minima have been carried out in their 1D profiles along the Os⋯N bond path. The QTAIM is based on the concept of atomic basins that are assigned

using the zero-flux condition [$\nabla\rho(r)\cdot n(r) = 0$] [38] in the electron density. The corresponding QTAIM surfaces are used to establish the interatomic boundaries. Equivalent boundaries can also be determined by using the electrostatic potential instead of the electron density, namely $\nabla\varphi(r)\cdot n(r) = 0$ [39], which establishes the bonded electroneutral centers. The resulting difference in both boundaries is useful to reveal the donor–acceptor nature of the interaction. In any interaction, the $\varphi(r)_{min}$ is shifted toward the electron donor atom, while the $\rho(r)_{min}$ is shifted toward the σ-hole donor atom (electron acceptor) [40–42].

The 1D profile of the ED and ESP functions along the Os(VI)···N bond path is shown in Figure 5. It can be observed that the Os(VI) atom acts as the electron acceptor since an evident shift of the ED minima toward the Os(VI) electron density basin is appreciated (see small arrow in Figure 5). This evidences that the N atom of pyridine is partially donating electrons to the electrophilic σ–hole of the Os(VI) metal center. This result is in line with the MEP analysis that shows a positive MEP at the Os(VI) atom in spite of the negative charge at the $[OsNCl_4]^-$ unit.

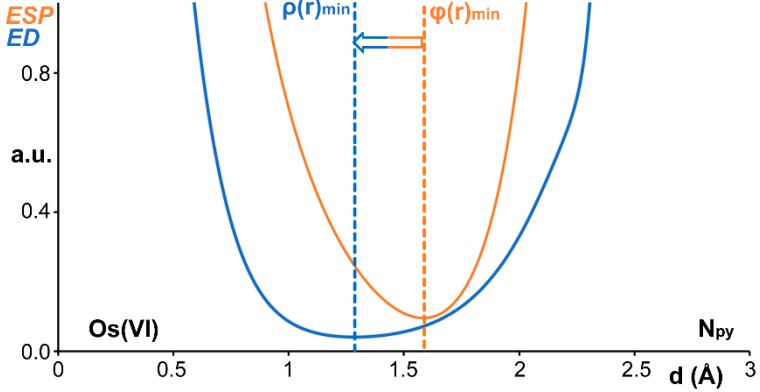

**Figure 5.** The ED (blue curve) and ESP (orange curve) 1D profile along the Os(VI)···N for complex **4**.

## 3. Computational Methods

The geometric optimizations were carried out using the Turbomole 7.2 program [43]. The level of theory used for these optimizations was the PBE0-D3/def2-TZVP [44–46]. For osmium, the def2-TZVP basis set used in this work includes effective core potentials (ECP), and relativistic effects are used for the inner electrons [46]. This combination of functional and basis sets has been successfully used before to study osme bonds [29,30]. The MEP surface plots are computed at the same level of theory and at the 0.002 a.u. isosurface. The analysis of the electron charge density was performed using the quantum theory of atoms in molecules (QTAIM) [47] and complemented by the noncovalent interCartesian Coordinates action plot index (NCIplot) [48] by using the reduced density gradient (RDG) isosurfaces. They were plotted using the VMD program [49]. The settings for the RDG plots were as follows: $s = 0.5$ a.u.; cut-off $\rho = 0.05$ a.u.; color scale $-0.04$ a.u. $\leq$ sign($\lambda_2$)$\rho \leq 0.04$ a.u. The natural bond orbital (NBO) [50] analysis was performed using the NBO7.0 program [51] at the PBE0-D3/def2-TZVP level of theory (Cartesian Coordinates in Supplementary Material).

## 4. Conclusions

The combined crystallographic and theoretical results reported in this work consistently prove the attractive interaction between nitrido-osmium(VI) complexes and lone pair donor atoms as an unprecedented σ-hole osme bond interaction. In several crystal structures, it is evidenced that the osme bond has a prominent role in determining the X-ray packing. They are particularly relevant in the formation of self-assembled dimers in FEJZUP and in the cocrystal of pyrazine and tetrachloro-nitrido-osmium(vi), governing the formation of 1:2 adducts (JOMKAX). Finally, the existence of a σ-hole opposite to the N≡Os bond and the participation of the antibonding σ*(N–Os) orbital in the interactions

have been also evidenced using the MEP surface and NBO analyses, respectively, thus, strongly supporting the σ-hole nature of the OmBs reported herein.

**Supplementary Materials:** The following supporting information can be downloaded at: https://www.mdpi.com/article/10.3390/inorganics10090133/s1, Cartesian coordinates of the optimized compounds.

**Author Contributions:** Conceptualization, R.M.G. and A.F.; methodology, R.M.G. and A.F.; formal analysis, R.M.G. and A.F.; investigation, R.M.G. and A.F.; writing—original draft preparation, A.F.; writing—review and editing, R.M.G. and A.F.; visualization, R.M.G. and A.F.; supervision, A.F.; project administration, A.F.; funding acquisition, A.F. All authors have read and agreed to the published version of the manuscript.

**Funding:** This research was funded by the MICIU/AEI of Spain (project PID2020-115637GB-I00 FEDER funds).

**Informed Consent Statement:** Not applicable.

**Data Availability Statement:** Not applicable.

**Acknowledgments:** We thank the Centre de Tecnologies de la Informació (CTI) at University of the Balearic Islands (UIB) for the computational facilities.

**Conflicts of Interest:** The authors declare no conflict of interest.

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
