# Peer review of "Crystallographic and Theoretical Study of Osme Bonds in Nitrido-Osmium(VI) Complexes"

_inorganics, doi:10.3390/inorganics10090133_

Round 1

Reviewer 1 Report

The paper of Antonio Frontera Rosa M. Gomila is an interesting fundamental work on searching non-covalent interaction of Os(VI) and lone pair donor atoms. This manuscript is a continuation of a previously published article “Molecular Electrostatic Potential and Noncovalent Interactions in Derivatives of Group 8 Elements”. I liked the original paper. In the current manuscript, which is a purely theoretical work, I am attracted by the method of researching previously published compounds. Previously, such databases like CCDC and computing power have not been available to chemists. The current work seems nice and interesting to me and I recommend it to publish in Inorganics.

The manuscript is written neatly, I have no significant comments on the content of the work. I hope this article will be accepted for publication and will be discussed by the scientific community.

Line 72 “teractions”

Line 116 “Mooreover”

Line 224 “anf”

Author Response

First of all, we would like to thank this reviewer for his/her careful reading of the manuscript, corrections and suggestions. Our replies follow:

Line 72 “teractions”

Reply: Fixed (the missing "in-" part is in the previous line)

Line 116 “Mooreover”

Reply: Fixed

Line 224 “anf”

Reply: Fixed

Reviewer 2 Report

The topic of this following article presented by Gomila et al depicts an interesting area of non conventional metal-ligand interactions of coordination chemistry. Although the area is new and interesting but the significance of this current work is not quite clear to understand. And how this study can be applied to solve the real research questions in this area and what is the takeaway has not been demonstrated clearly. Also there are many typos/ grammatical mistakes during the construction of the manuscript which should be taken care of so that it can interest to broad spectrum of readers. 

Author Response

First of all, we would like to thank this reviewer for his/her careful reading of the manuscript, corrections and suggestions. Our replies follow:

1) The manuscript is devoted to the study of Osme bonds in nitrido-osmium(VI) complexes, which is unprecedented. The research reported in the manuscript is quite fundamental, so there is not an immediate application for this research in terms of solving real experimental problems.

2) We have eliminated some typos and revised the grammar.

Reviewer 3 Report

This is an interesting manuscript describing the sigma hole interactions in some nitrido-osmium(VI) complexes. The authors have established crystal structures of four novel complexes possessing osme bonds and supported the experimental data with computational calculations. Though osme bonds in OsO4 are previously reported by the same group (ref 29), the present study emphasizes the osme interactions in Os(VI) compounds. The study has been carefully done and the results are scientifically presented. Therefore, I recommend publication of this work in inorganics. Prior to publication, I recommend a careful review of the manuscript to refine the grammar, and spell check is done throughout the manuscript.

Minor points:-

Page 1, line 8: any atom of group of atoms…. This should be “any atom or group of atoms.”

Introduction line 19: aerogen,….should be “argon”.

Line 21: rationalized used the σ-hole and π-hole concepts propose…. Should be “rationalized as σ-hole and π-hole concepts”…

Line 41: exhibit interesting photochemistry [32-34] 41 and electrochemistry [35,36] properties.. should be “exhibit interesting photochemical [32-34] 41 and electrochemical [35,36] properties.”

Line 70: figure 2b.. should be “figure 1b”

Author Response

First of all, we would like to thank this reviewer for his/her careful reading of the manuscript, corrections and suggestions. Our replies follow:

Page 1, line 8: any atom of group of atoms…. This should be “any atom or group of atoms.”

Reply: Fixed, thanks

Introduction line 19: aerogen,….should be “argon”.

Reply: This term is correct, aerogen term has been used to define any atom of group 18.

Line 21: rationalized used the σ-hole and π-hole concepts propose…. Should be “rationalized as σ-hole and π-hole concepts”…

Reply: Fixed, thanks

Line 41: exhibit interesting photochemistry [32-34] 41 and electrochemistry [35,36] properties.. should be “exhibit interesting photochemical [32-34] 41 and electrochemical [35,36] properties.”

Reply: Fixed, thanks

Reviewer 4 Report

The authors introduce a new term "Osme bond", which is not appropriate, because it does not refer to a unique phenomenon. The weakening of the coordination bond in the trans position to the covalent bond is a fairly common phenomenon. The fact that the multiple bonds weaken the trans position even more is also wide known.
The presentation of coordination bonds as donor-acceptor ("electron pair" + "hole") in the method of valence bonds is included in all textbooks on inorganic chemistry.
Table 1 in the peer-reviewed article does not contain important information. An indication of the number of structures and structural formulas of all the compounds mentioned in the article is sufficient.

Summing up, the article does not contain fundamentally new information, but is interesting from the point of view of quantum-chemical calculations characterizing the structure of the osmium
(VI) complex. The article can be published.

Author Response

First of all, we would like to thank this reviewer for his/her careful reading of the manuscript, corrections and suggestions. Our replies follow:

1) The Osme bond it not a new term, it has been used before, see references 29 and 30.

2) If the referee agrees, we would prefer to keep Table 1 in the main text, since the potential readers will have access to all structures in the CSD with the Os≡N bonds

Round 2

Reviewer 2 Report

I appreciate the authors for the clarificaion and modificaion of the manuscript. The work can be published.